# OphthoEvidence report: Baseline structural optical coherence tomography biomarkers predictive of visual acuity following epiretinal membrane surgery–A systematic review and meta-analysis protocol

Xiajing Chu[1,2], João Pedro Lima[1,2,3], Dena Zeraatkar[1,2,3], Jinhui Ma[1,3], Femin Prasad[2], Sangeetha Srinivasan[4], Sobha Sivaprasad[5], Teresa Sandinha[6], Peter Kaiser[7], David H. Steel[8,9], Tien Yin Wong[10,11], Charles C. Wykoff[12], Varun Chaudhary[1,2,3]*

1 Department of Health Research Methods, Evidence, and Impact, McMaster University, Hamilton, Ontario, Canada, 2 Division of Ophthalmology, Department of Surgery, McMaster University, Hamilton, Ontario, Canada, 3 Thomas Kevill Vitreoretinal Imaging Lab, St. Joseph's Healthcare, Hamilton, Ontario, Canada, 4 Division of Ophthalmology, Department of Surgery, St. Joseph's Health Care, Hamilton, Ontario, Canada, 5 NIHR Biomedical Research Centre at Moorfields Eye Hospital and UCL Institute of Ophthalmology, London, United Kingdom, 6 St Paul's Eye Unit, Royal Liverpool and Broadgreen University Hospitals NHS Trust, Liverpool, United Kingdom, 7 Cole Eye Institute, Cleveland Clinic, Cleveland, Ohio, 8 Bioscience Institute, Newcastle University, Newcastle Upon Tyne, United Kingdom, 9 Sunderland Eye Infirmary, Sunderland, United Kingdom, 10 Singapore Eye Research Institute, Singapore National Eye Centre, Singapore, Singapore, 11 Beijing Visual Science and Translational Eye Research Institute, School of Clinical Medicine, Beijing Tsinghua Changgung Hospital, Tsinghua Medicine, Tsinghua University, Beijing, China, 12 Retina Consultants of Texas, Houston, Texas, Blanton Eye Institute, Houston Methodist Hospital, Houston, Texas, United States of America

* vchaudh@mcmaster.ca

## Abstract

### Background

Epiretinal membrane is a common retinal condition, particularly in the elderly, than can lead to reduced visual acuity and visual distortion. Structural optical coherence tomography biomarkers have shown potential for predicting visual outcomes following surgery.

### Objective

To evaluate the prognostic value of baseline structural optical coherence tomography biomarkers for visual outcomes following epiretinal membrane surgery. We will assess associations between individual biomarkers and postoperative visual acuity or changes in visual acuity at 6, 12, and 24 months after surgery.

### Methods

We will systematically search MEDLINE, EMBASE, CENTRAL and Web of Science from inception. Eligible studies will include randomized trials and observational

**Data availability statement:** Deidentified research data will be made publicly available when the study is completed and published.

**Funding:** The author(s) received no specific funding for this work.

**Competing interests:** All authors have completed and submitted the ICMJE disclosures form. XC, JPL, DZ, JM: None; SS reports grants from Bayer, Roche, Boehringer Ingelheim, and AbbVie; consultancy for AbbVie, Alimera Sciences, Amgen, Apellis, Astellas, Bayer, Biogen, Boehringer Ingelheim, Clearside Biomedical, Eyebiotech, EyePoint Pharmaceuticals, Iveric Bio, Janssen Pharmaceuticals, Novo Nordisk, Optos, Ocular Therapeutix, Kriya Therapeutics, OcuTerra, Ripple Therapeutics, Roche, Stealth Biotherapeutics, and Sanofi; payment or honoraria from Bayer, Roche, Boehringer Ingelheim, and AbbVie; meeting/travel support from Bayer and Roche; participation on an advisory board or DSMB for Novo Nordisk; stock or stock options in EyeBiotech; and receipt of materials or services from Boehringer Ingelheim. DHS: Consultant for Alcon, BVI, DORC, Roche, Alimera, Eyepoint, Complement therapeutics, Sitala, AviadoBio; Research funding: Alcon, Bayer, DORC, BVI, Boehringer-Ingelheim; PK: Dr. Kaiser serves as a consultant for AAVAntgarde Bio, Abbvie, Alcon, Alexion, Alkeus, Allgenesis, Alzheon, Amaros, Annexon Biosciences, AsclepiX, Astellas, Augen Therapeutics, Aviceda, Bayer, Bausch and Lomb, Beacon Therapeutics (AGTC), Biogen Idec, Bionic Vision Technologies, Carl Zeiss Meditec, Celltrion Healthcare Co., Complement Therapeutics, Endogena Therapeutics, Frontera Therapeutics, Galimedix, Innovent, Invirsa, iRenix, Isarna, Janssen, jCyte, Kanaph Therapeutics, Kanghong, Kera Therapeutics, Kriya Therapeutics, Nanoscope Therapeutics, Ocugenix, Oculis, Omeros, Osanni Bio, Panther Pharmaceuticals, Ray Therapeutics, RegenxBio, Resonance Medicine Inc., RetinaAI Medical AG, Retinal Sciences, ReVana, Revopsis, Roivant, Samsung Bioepis, Sandoz, SGN Nanopharma Inc., SmileBiotek Zhuhai Ltd, Stealth Biotherapeutics, Stuart, Sustained Nano Systems, Takeda, Théa, Tilak, Unity Biotechnology, Vanotech, VisgenX; He

studies reporting the association between any baseline optical coherence tomography biomarkers and postoperative visual acuity in patients undergoing epiretinal membrane surgery. Two reviewers will independently and in duplicate perform screening, data extraction, and risk of bias assessment. Meta-analyses using the restricted maximum likelihood random-effects model will be conducted whenever possible. The certainty of evidence for each estimate will be assessed using the GRADE approach.

## Expected outcomes

This review will analyze time-specific association between baseline structural optical coherence tomography biomarkers and postoperative visual acuity and change in visual acuity from baseline at 6, 12, and 24 months after surgery, quality of life measured using any validated scale, and adverse events.

---

## Introduction

Epiretinal membrane (ERM) is a sheet-like layer of fibrous tissue that forms on the inner surface of the retina through the proliferation of myofibroblast cells and associated extracellular matrix components [1]. Its prevalence in the adult population is about 9%, rising to nearly 20% among those over 75, with no clear sex differences [2]. ERM is classified as idiopathic, usually age-related, or secondary, occurring in the context of pre-existing ocular disease, trauma, or prior retinal interventions [1,3,4].

Patients with ERM may experience reduced visual acuity (VA), metamorphopsia, micropsia, or monocular diplopia [5]. The primary treatment goals are to preserve or improve vision [1,3]. Surgery is generally reserved for patients with significant impairment or disturbing distortion, with pars plana vitrectomy and membrane peeling serving as the standard intervention [6,7].

Currently, there is no clear guidance for surgical indications [1]. The American Association of Ophthalmology (AAO), for example, recommends that surgery should be performed based on patient's discomfort level while balancing the risks of a surgical procedure [8]. A literature review only found low to very low certainty of evidence from one randomized trial assessing the effectiveness of surgery for patients with ERM [9]. This lack of evidence and guidelines results in variability in practice and clinical outcomes. The identification of optical coherence tomography (OCT) biomarkers associated with better visual outcomes following surgery may help balance the potential benefits and risks of surgery and, consequently, optimize personalized surgical decisions [10].

OCT, a non-invasive imaging modality central to the diagnosis and monitoring of macular disease, provides both qualitative and quantitative assessment of macular microstructure [11–15]. Structural OCT biomarkers have shown promise in predicting visual outcomes and monitoring disease progression [16]. However, inconsistencies in study design and methodological quality hinder the integration of OCT biomarkers

is an employee of Ocular Therapeutics. TYW: Professor Wong is a consultant for Astellas, Bayer, Boehringer-Ingelheim, Genentech, Iveric Bio, Novartis, Oxurion, Plano, Roche, Sanofi, and Shanghai Henlius. He is an inventor, holds patents and is a co-founder of start-up companies EyRiS and Visre, which have interests in, and develop digital solutions for eye diseases; CCW: Dr. Wykoff reports consulting for 4DMT, AbbVie, Adverum, Alcon, Alimera, Alkeus, Annexon, Apellis, Aviceda, Bayer, Biocryst, Boehringer Ingelheim, Clearside, EyeBiotech, EyePoint, Genentech, InGel, Janssen, Kiora, KodiakMerck, Neurotech, Novartis, Ocuphire, ONL, Opthea, Osanni, Panther, Perceive Bio, Ray, Regeneron, RegenXBio, Sanofi, Santen, Stealth, Valo, Zeiss.; VC: Dr. Chaudhary reports acting as an advisory board member, grants and other from Novartis; acting as an advisory board member, grants and other from Bayer; grants from Allergan; acting as an advisory board member and grants from Roche; acting as an advisory board member for Janssen; acting as an advisory board member for Apellis; acting as an advisory board member for Boehringer Ingelheim, and acting as an advisory board member for EyePoint outside the submitted work. There are no patents, products in development or marketed products associated with this research to declare. This does not alter our adherence to PLOS ONE policies on sharing data and materials.

into clinical practice. A robust, comprehensive review of the evidence supporting the association between OCT biomarkers and visual outcomes is still needed.

This systematic review and meta-analysis aims to assess the prognostic value of individual baseline OCT biomarkers for visual outcomes following ERM surgery using a comprehensive and rigorous methodology [17]. We will investigate the association between OCT biomarkers and changes in VA at 6 months, 12 months and 24 months after treatment initiation. By summarizing the existing evidence, this review seeks to facilitate more personalized treatment approaches and deepen our understanding of prognostic factors in ERM care.

## Materials and methods

We adhere to the 2015 PRISMA-P (Preferred Reporting Items for Systematic reviews and Meta-Analyses extension for protocols) statement (S1 file) [18]. We register this protocol on Open Science Framework (https://osf.io).

### Eligibility criteria

This systematic review will include data from randomized controlled trials, cohort studies, and case series that meet all of the following criteria: 1) involve patients with ERM undergoing surgery, 2) assess OCT biomarkers measured before surgery (ideally within one year prior to surgery) to determine their predictive value for treatment response, either individually or as part of a multivariable clinical prediction model.

Criteria for exclusion are: 1) Studies including secondary ERMs associated with other ocular conditions (e.g., diabetic macular edema, retinal vein occlusion, rhegmatogenous retinal detachment, uveitis, or fibrovascular proliferation); 2) Studies including eyes with full-thickness macular hole; 3) Studies including post-pars plana vitrectomy ERMs; 4) studies including eyes with full-thickness macular hole (partial-thickness tractional holes, epiretinal membrane foveoschisis, and lamellar macular holes will be included) or with advanced macular co-pathology affecting vision; 5) studies with less than 10 patients with ERM; 6) studies describing techniques or guidelines; 7) studies in which surgical parameters are the primary measures of outcome; 8) studies with no stratified analysis for patients with ERM; 9) non-original research/review.

We are interested in the following outcomes at 6 months, 12 months, or 24 months after surgery: [9]:

- Final postoperative best-corrected visual acuity (BCVA) and mean change in BCVA, measured using a standardized chart (e.g., Logarithm of the Minimum Angle of Resolution (logMAR), Snellen) at a starting distance of 4 m.

- Proportion of patients with a gain of>= 0.2 logMAR in BCVA or uncorrected visual acuity (UCVA) in the study eye.

- Proportion of patients with a loss of>=0.2 logMAR in BCVA or UCVA in the study eye.

- Mean change in quality of life (QoL), measured using a validated questionnaire.

- Any adverse events (AEs) identified during follow-up.

A threshold of 0.2 logMAR (10 Early Treatment Diabetic Retinopathy Study (ETDRS) letters or 2 Snellen lines) will be selected because this value is recognized in ophthalmology research [9] and has been adopted in clinical trial of retinal diseases [19]. We will search without restrictions on language or date of publication.

### Search strategy

With support from an experienced research librarian, we will search Ovid MEDLINE, EMBASE, CENTRAL and Web of Science using a combination of keywords and Medical Subject Headings (MeSH) terms. We will systematically search from inception. We will supplement our search with backward and forward citation searching using *citationchaser* (https://estech.shinyapps.io/citationchaser/) and search the reference lists of systematic reviews identified [9,16,20]. To ensure inclusion of all relevant studies, we will contact clinical experts in the field.

### Screening

After training and calibration exercises, reviewers will work independently and in duplicate to screen titles and abstracts of search records and, subsequently, full texts to determine eligibility. All disagreements will be resolved by discussion and, when necessary, by a third reviewer. We will use Covidence (Melbourne, Australia, https://www.covidence.org), an online systematic review software for screening titles and abstracts and full-text articles. A PRISMA Flow Diagram will provide a visual overview of the records identified, included and excluded, and the reasons for exclusion [21].

### Data extraction

Two reviewers, working independently and in duplicate, will collect data using a standardized pilot-tested collection form designed in Microsoft Excel (Version 16.52). Discrepancies in data collection will be resolved by discussion and disagreements will be resolved with a third reviewer. We will contact the corresponding author or the sponsoring pharmaceutical company for clarification in case of missing data.

For each included study, we will extract study characteristics, patient characteristics, surgical techniques, biomarker(s) examined and their definitions, whether an image reading center was employed to identify biomarkers, the association between baseline biomarker(s) and visual outcomes of interest (we listed the detailed items in Table 1). We listed the potential biomarkers and definitions in Table 2. Other structural OCT biomarkers not listed in Table 2 will also be eligible for inclusion if reported.

We will extract VA, QoL, and AEs at 6 months (±3 months), 12 months (±3 months) and 24 months (±3 months) after treatment initiation, type of analysis conducted, whether the analysis is univariate or multivariate. If multivariate, we will extract the variables controlled for in the analysis and whether the study follows the Transparent Reporting of a multivariable prediction model for Individual Prognosis or Diagnosis (TRIPOD) guidelines [22]. We will extract whether bilateral eyes are allowed, and if so, how they are handled in the analysis.

### Risk of bias assessment

We will use the Quality in Prognosis Studies (QUIPS) tool to assess risk of bias for prognostic factor studies [23]. For studies that develop or validate multivariable clinical prediction models, we will apply the Prediction model Risk Of Bias ASsessment tool (PROBAST) [24]. QUIPS tool assesses risk of bias across six domains: participation, attrition, prognostic factor measurement, confounding measurement and account, outcome measurement, and statistical analysis and reporting. There are four domains (participants, predictors, outcome, and analysis), and total of 20 signaling questions in

**Table 1. Data extraction items.**

| Category | Items to be extracted |
|---|---|
| Study characteristics | Author, publication year, study design, funding. |
| Patient characteristics | Age, sex, lens status (both pre- and post-operative), visual acuity, Type of ERM (idiopathic or secondary), comorbidities (if available). |
| Surgical techniques | Pars plana vitrectomy alone vs. combined phacoemulsification and pars plana vitrectomy, gauge size, internal limiting membrane peel, type of dye, type of tamponade, adjunctive or concomitant treatment strategies (e.g., peri-/post-operative non-steroidal anti-inflammatory drug or steroid. |
| Biomarkers | Baseline OCT biomarker (s) examined and their definitions (see Table 2 for detailed biomarker definitions). |
| Image assessment | Whether an image reading center is employed to identify biomarkers. |
| Bilateral eyes | Whether bilateral eyes are allowed, and if so, how they are handled in the analysis. |
| Outcomes of interest | Final postoperative BCVA and mean change in BCVA, QoL, AEs. |
| Effect size | Association between baseline biomarker(s) and visual outcomes of interest. |

**Table 2. The list of potential baseline structural OCT biomarkers and definitions.**

| Biomarker | Definition |
|---|---|
| Central foveal thickness | Thickness of the retina at the foveal center. |
| Inner segment/outer segment integrity | Continuity of the ellipsoid zone band on OCT. |
| Cone outer segment tips integrity | Continuity of the hyperreflective line external to the cone outer segment tips junction. |
| Disorganization of the retinal inner layers | Inability to distinguish boundaries between ganglion cell layer, inner plexiform layer, and inner nuclear layer. |
| External limiting membrane integrity | Continuity of the hyperreflective external limiting membrane band. |
| Presence of ectopic inner foveal layers | Extension of the inner nuclear layer and inner plexiform layer across the fovea. |
| ERM staging (Govetto classification) | Morphological classification of ERM on OCT. |
| Foveal contour | Shape of the fovea on OCT. |
| Retinal pigment epithelium integrity | Continuity of the retinal pigment epithelium layer. |
| Cystoid macular edema | Presence of cystoid spaces within the retina. |
| Central Bouchet changes (Cotton ball sign, subretinal fluid, vitelliform lesions) | Presence of central hyperreflective or subretinal changes associated with ERM traction. |
| Interface changes (vitreomacular Traction, scrolling, nerve Fiber Layer schisis) | Alterations at the vitreoretinal interface due to ERM traction. |
| Foveal ectopia and post-op movement | Displacement of the foveal center and its change after surgery. |

PROBAST. Any disagreements will be resolved through discussion. For the risk of study confounding to be rated as low, we will pre-specify that the study must report a multivariable model that includes, at a minimum, age [20], patients' baseline VA [16] and baseline retinal thickness [25]. The overall risk of bias will be deemed low if all domains are rated as low

risk, and high if one or more domains are rated as high risk. Otherwise, the study will be rated as having a moderate risk of bias.

## Data analysis

We will conduct meta-analyses using the restricted maximum likelihood random-effects model for all prognostic factors reported by two or more studies. VA is frequently reported using the ETDRS letter score. We will convert values reported in Logarithm of the logMAR, Snellen chart scores or any other validated scale to approximate ETDRS scores using the method proposed by Gregori et al [26]. If multiple scales measured the same outcome, we used linear transformation to the most used scale [27]. For dichotomous outcomes, we will convert relative effects into absolute effects to facilitate interpretation and magnitude of effect. We will calculate absolute effects using pooled RRs applying the median risk in control groups of studies that report 1 or more events as the baseline risk. If both eyes from the same patient are included in a study and analyzed as independent units without appropriate adjustment, we will consider this in the risk of bias assessment and explore the potential impact in sensitivity analyses. Where possible, we will extract effect estimates that account for within-patient correlation; otherwise, we may include only one eye per patient (e.g., randomly selected or as reported by study authors) to avoid double-counting.

If the same OCT biomarker is labeled differently across studies, we will extract each study's definition and group those that appear sufficiently similar. To minimize bias, the definitions will be shared with three ophthalmologists blinded to study results, and they will confirm or revise our grouping decisions. When consensus cannot be reached, or definitions differ substantially, results will be summarized narratively. All analyses will be performed in R (Version 4.3.1) using the *meta* and *metafor* packages [28,29]. When quantitative analysis is not possible, we will present results narratively using the SWiM (Synthesis Without Meta-analysis) guidance [30]

If two or more studies are available for any subgroup, the following subgroup analyses will be conducted:

i)   High vs. low risk of bias, hypothesizing that studies with higher risk of bias may overestimate the strength of associations between baseline OCT biomarkers and visual outcomes.

ii)  Presence of a reading center, hypothesizing that studies not using a reading center may overestimate the associations [31,32].

iii) Lens status and surgery type (pars plana vitrectomy alone vs. combined phacoemulsification and pars plana vitrectomy), hypothesizing that lens removal independently improves VA, and phakic eyes tend to develop or progress cataract after vitrectomy.

iv)  Internal limiting membrane peel vs. no peel, hypothesizing that internal limiting membrane peeling may reduce recurrence.

v)   Govetto stage/ presence of ectopic inner foveal layers, hypothesizing that greater disease indicates chronic traction and neuroglial remodeling, which may limit postoperative visual recovery.

vi)  Partial-thickness macular defects: presence vs. absence of partial thickness holes (partial-thickness tractional holes, epiretinal membrane foveoschisis, and lamellar macular holes). We hypothesize that eyes with these conditions will show less visual improvement after ERM surgery compared with eyes without such defects.

vii) Degree of adjustment for critical predictors of outcome, hypothesizing that models that adequately adjust for key predictors may yield more reliable effect estimates.

The credibility of the statistically significant subgroup effects will be assessed using the Instrument to assess the Credibility of Effect Modification Analyses (ICEMAN) [33]. We will only present stratified results if moderate to high credibility subgroup effects are found.

Publication bias will be assessed by visually inspecting the funnel plots for outcomes reported in ten or more studies. If less than ten studies are included for an outcome, publication bias will be assessed by evaluating the quality of the search and measures taken to obtain all possible evidence to inform the outcome of interest [34].

## Certainty of evidence

The certainty of evidence for this systematic review will be evaluated using the Grading of Recommendations, Assessment, Development, and Evaluation (GRADE) approach for prognosis [35,36], based on several key factors: methodological limitations, indirectness, imprecision, inconsistency, and potential publication bias [36]. The ratings will be determined using a minimal clinically important difference (MCID) of 15 ETDRS letters [37,38]. For dichotomous outcomes, we will use an absolute risk difference of 10% as the MCID.

The GRADE ratings will provide an overall confidence level in the evidence. A "High" rating will indicate that the true effect is very likely close to the estimated effect. A "Moderate" rating will suggest that the true effect is probably similar to the reported estimate, while a "Low" rating will imply that the true effect may be substantially different. A "Very Low" rating will reflect a high degree of uncertainty, limiting the ability to draw firm conclusions. These ratings will be used to determine whether each biomarker is associated with improved, stable, or worsened VA. We will use standardized language that incorporates certainty of evidence when communicating our results (i.e., high certainty evidence presented with declarative statements, moderate certainty evidence with 'likely', low certainty evidence with 'may', and very low indicated by 'very uncertain') [39].

## Discussion

This systematic review and meta-analysis will synthesize the available evidence on baseline structural OCT biomarkers and their prognostic value for visual outcomes following ERM surgery. Using rigorous methods – including duplicate screening and data collection, standardized outcome definitions, risk of bias assessment, and evaluation of certainty of evidence – our review will provide a clearer understanding of which OCT-derived features are most predictive of postoperative VA. Ultimately, these findings may inform the development of individualized prognostic tools to support shared decision-making and improve clinical management in patients with idiopathic ERM.

Visual outcomes after ERM surgery vary widely between patients, and predicting these outcomes remains a key challenge in clinical practice. Numerous studies have investigated the prognostic value of preoperative factors, particularly those measured by structural OCT, including inner segment/outer segment integrity, severity of metamorphopsia, cone outer segment tips integrity, and fundus autofluorescence, to identify reliable predictors of postoperative VA [16,40–42]. Despite these findings, the current body of evidence is limited by methodological heterogeneity, inconsistent reporting of effect sizes, and a lack of multivariable prognostic models accounting for all critical confounding factors.

To our knowledge, this study will be the first meta-analysis to quantitatively evaluate the prognostic value of baseline structural OCT biomarkers for visual outcomes following ERM surgery. Several potential limitations from included evidence are anticipated. First, there may be inconsistency in the selection of OCT biomarkers across studies, which may contribute to heterogeneity in the pooled estimates. Second, variations in statistical approaches (univariate versus multivariate analyses) and differences in the covariates included in multivariable models may limit the comparability of effect estimates. Third, the follow-up durations and visual outcome assessments time may not be strictly defined. Fourth, the nature of observational studies may limit causal inference. Finally, patient populations may differ in disease severity, and etiology (idiopathic versus secondary ERM), potentially affecting the generalizability of the findings, which we will address by conducting subgroup analyses whenever possible.

This review and have several strengths. First, it will employ rigorous and transparent methods in accordance with PRISMA-P guidelines, including a comprehensive search strategy and duplicate screening, data extraction, and risk of bias assessment. Second, we will apply validated tools to assess the risk of bias and the certainty of evidence, offering a

structured and transparent summary of the strength of each prognostic association. Third, by analyzing visual outcomes at clinically relevant time points (6, 12, and 24 months), this review will provide time-specific estimates of prognostic value.

The findings of this review will guide future research, support clinical decision-making, and lay the groundwork for developing individualized prediction models and clinical practice guidelines in ERM management.

## Supporting information

**S1 File. PRISMA-P sheets.**
(DOCX)

## Author contributions

**Conceptualization:** Xiajing Chu, João Pedro Lima, Dena Zeraatkar.

**Data curation:** Xiajing Chu, João Pedro Lima.

**Formal analysis:** Xiajing Chu, João Pedro Lima.

**Funding acquisition:** Varun Chaudhary.

**Investigation:** Xiajing Chu, João Pedro Lima.

**Methodology:** Xiajing Chu, João Pedro Lima, Dena Zeraatkar, Jinhui Ma, Femin Prasad, Sangeetha Srinivasan, Varun Chaudhary.

**Project administration:** Xiajing Chu, João Pedro Lima, Dena Zeraatkar, Varun Chaudhary.

**Resources:** Xiajing Chu, Femin Prasad, Sangeetha Srinivasan, Sobha Sivaprasad, Teresa Sandinha, Peter Kaiser, David H. Steel, Tien Yin Wong, Charles C. Wykoff, Varun Chaudhary.

**Software:** Xiajing Chu, João Pedro Lima.

**Supervision:** João Pedro Lima, Dena Zeraatkar, Jinhui Ma, Femin Prasad, Sangeetha Srinivasan, Sobha Sivaprasad, Teresa Sandinha, Peter Kaiser, David H. Steel, Tien Yin Wong, Charles C. Wykoff, Varun Chaudhary.

**Validation:** Xiajing Chu, João Pedro Lima, Dena Zeraatkar.

**Visualization:** Xiajing Chu, João Pedro Lima.

**Writing – original draft:** Xiajing Chu.

**Writing – review & editing:** Xiajing Chu, João Pedro Lima, Dena Zeraatkar, Jinhui Ma, Femin Prasad, Sangeetha Srinivasan, Sobha Sivaprasad, Teresa Sandinha, Peter Kaiser, David H. Steel, Tien Yin Wong, Charles C. Wykoff, Varun Chaudhary.

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
