## [Editor Report · Decision Letter 0]

20 Nov 2025

OphthoEvidence Report: Baseline structural optical coherence tomography biomarkers predictive of visual acuity following epiretinal membrane surgery- A systematic review andmeta-analysis protocol

PONE-D-25-59787

Dear Dr. Chaudhary,

We’re pleased to inform you that your manuscript has been judged scientifically suitable for publication and will be formally accepted for publication once it meets all outstanding technical requirements.

Kind regards,

Andrzej Grzybowski

Academic Editor

PLOS ONE

Journal Requirements:

1. Thank you for stating the following in the Competing Interests section:

[All authors have completed and submitted the ICMJE disclosures form. XC, JPL, DZ, JM: None; SS reports grants from Bayer, Roche, Boehringer Ingelheim, and AbbVie; consultancy for AbbVie, Alimera Sciences, Amgen, Apellis, Astellas, Bayer, Biogen, Boehringer Ingelheim, Clearside Biomedical, Eyebiotech, EyePoint Pharmaceuticals, Iveric Bio, Janssen Pharmaceuticals, Novo Nordisk, Optos, Ocular Therapeutix, Kriya Therapeutics, OcuTerra, Ripple Therapeutics, Roche, Stealth Biotherapeutics, and Sanofi; payment or honoraria from Bayer, Roche, Boehringer Ingelheim, and AbbVie; meeting/travel support from Bayer and Roche; participation on an advisory board or DSMB for Novo Nordisk; stock or stock options in EyeBiotech; and receipt of materials or services from Boehringer Ingelheim. DHS: Consultant for Alcon, BVI, DORC, Roche, Alimera, Eyepoint, Complement therapeutics, Sitala, AviadoBio; Research funding: Alcon, Bayer, DORC, BVI, Boehringer-Ingelheim; PK: Dr. Kaiser serves as a consultant for AAVAntgarde Bio, Abbvie, Alcon, Alexion, Alkeus, Allgenesis, Alzheon, Amaros, Annexon Biosciences, AsclepiX, Astellas, Augen Therapeutics, Aviceda, Bayer, Bausch and Lomb, Beacon Therapeutics (AGTC), Biogen Idec, Bionic Vision Technologies, Carl Zeiss Meditec, Celltrion Healthcare Co., Complement Therapeutics, Endogena Therapeutics, Frontera Therapeutics, Galimedix, Innovent, Invirsa, iRenix, Isarna, Janssen, jCyte, Kanaph Therapeutics, Kanghong, Kera Therapeutics, Kriya Therapeutics, Nanoscope Therapeutics, Ocugenix, Oculis, Omeros, Osanni Bio, Panther Pharmaceuticals, Ray Therapeutics, RegenxBio, Resonance Medicine Inc., RetinaAI Medical AG, Retinal Sciences, ReVana, Revopsis, Roivant, Samsung Bioepis, Sandoz, SGN Nanopharma Inc., SmileBiotek Zhuhai Ltd, Stealth Biotherapeutics, Stuart, Sustained Nano Systems, Takeda, Théa, Tilak, Unity Biotechnology, Vanotech, VisgenX; He is an employee of Ocular Therapeutics. TYW: Professor Wong is a consultant for Astellas, Bayer, Boehringer-Ingelheim, Genentech, Iveric Bio, Novartis, Oxurion, Plano, Roche, Sanofi, and Shanghai Henlius. He is an inventor, holds patents and is a co-founder of start-up companies EyRiS and Visre, which have interests in, and develop digital solutions for eye diseases; CCW: Dr. Wykoff reports consulting for 4DMT, AbbVie, Adverum, Alcon, Alimera, Alkeus, Annexon, Apellis, Aviceda, Bayer, Biocryst, Boehringer Ingelheim, Clearside, EyeBiotech, EyePoint, Genentech, InGel, Janssen, Kiora, KodiakMerck, Neurotech, Novartis, Ocuphire, ONL, Opthea, Osanni, Panther, Perceive Bio, Ray, Regeneron, RegenXBio, Sanofi, Santen, Stealth, Valo, Zeiss.; VC: Dr. Chaudhary reports acting as an advisory board member, grants and other from Novartis; acting as an advisory board member, grants and other from Bayer; grants from Allergan; acting as an advisory board member and grants from Roche; acting as an advisory board member for Janssen; acting as an advisory board member for Apellis; acting as an advisory board member for Boehringer Ingelheim, and acting as an advisory board member for EyePoint outside the submitted work.]. 

We note that you received funding from a commercial source: Bayer, Roche, Boehringer Ingelheim, AbbVie, Alcon, DORC, BVI, Allergan,

Please respond by return email with your amended Competing Interests Statement and we will change the online submission form on your behalf.

[All authors have completed and submitted the ICMJE disclosures form. XC, JPL, DZ, JM: None; SS reports grants from Bayer, Roche, Boehringer Ingelheim, and AbbVie; consultancy for AbbVie, Alimera Sciences, Amgen, Apellis, Astellas, Bayer, Biogen, Boehringer Ingelheim, Clearside Biomedical, Eyebiotech, EyePoint Pharmaceuticals, Iveric Bio, Janssen Pharmaceuticals, Novo Nordisk, Optos, Ocular Therapeutix, Kriya Therapeutics, OcuTerra, Ripple Therapeutics, Roche, Stealth Biotherapeutics, and Sanofi; payment or honoraria from Bayer, Roche, Boehringer Ingelheim, and AbbVie; meeting/travel support from Bayer and Roche; participation on an advisory board or DSMB for Novo Nordisk; stock or stock options in EyeBiotech; and receipt of materials or services from Boehringer Ingelheim. DHS: Consultant for Alcon, BVI, DORC, Roche, Alimera, Eyepoint, Complement therapeutics, Sitala, AviadoBio; Research funding: Alcon, Bayer, DORC, BVI, Boehringer-Ingelheim; PK: Dr. Kaiser serves as a consultant for AAVAntgarde Bio, Abbvie, Alcon, Alexion, Alkeus, Allgenesis, Alzheon, Amaros, Annexon Biosciences, AsclepiX, Astellas, Augen Therapeutics, Aviceda, Bayer, Bausch and Lomb, Beacon Therapeutics (AGTC), Biogen Idec, Bionic Vision Technologies, Carl Zeiss Meditec, Celltrion Healthcare Co., Complement Therapeutics, Endogena Therapeutics, Frontera Therapeutics, Galimedix, Innovent, Invirsa, iRenix, Isarna, Janssen, jCyte, Kanaph Therapeutics, Kanghong, Kera Therapeutics, Kriya Therapeutics, Nanoscope Therapeutics, Ocugenix, Oculis, Omeros, Osanni Bio, Panther Pharmaceuticals, Ray Therapeutics, RegenxBio, Resonance Medicine Inc., RetinaAI Medical AG, Retinal Sciences, ReVana, Revopsis, Roivant, Samsung Bioepis, Sandoz, SGN Nanopharma Inc., SmileBiotek Zhuhai Ltd, Stealth Biotherapeutics, Stuart, Sustained Nano Systems, Takeda, Théa, Tilak, Unity Biotechnology, Vanotech, VisgenX; He is an employee of Ocular Therapeutics. TYW: Professor Wong is a consultant for Astellas, Bayer, Boehringer-Ingelheim, Genentech, Iveric Bio, Novartis, Oxurion, Plano, Roche, Sanofi, and Shanghai Henlius. He is an inventor, holds patents and is a co-founder of start-up companies EyRiS and Visre, which have interests in, and develop digital solutions for eye diseases; CCW: Dr. Wykoff reports consulting for 4DMT, AbbVie, Adverum, Alcon, Alimera, Alkeus, Annexon, Apellis, Aviceda, Bayer, Biocryst, Boehringer Ingelheim, Clearside, EyeBiotech, EyePoint, Genentech, InGel, Janssen, Kiora, KodiakMerck, Neurotech, Novartis, Ocuphire, ONL, Opthea, Osanni, Panther, Perceive Bio, Ray, Regeneron, RegenXBio, Sanofi, Santen, Stealth, Valo, Zeiss.; VC: Dr. Chaudhary reports acting as an advisory board member, grants and other from Novartis; acting as an advisory board member, grants and other from Bayer; grants from Allergan; acting as an advisory board member and grants from Roche; acting as an advisory board member for Janssen; acting as an advisory board member for Apellis; acting as an advisory board member for Boehringer Ingelheim, and acting as an advisory board member for EyePoint outside the submitted work.].   

We note that one or more of the authors are employed/affiliated by a commercial company: 4DMT, AAVAntgarde Bio, AbbVie, Adverum, Alcon, Alexion, Alimera Sciences, Alkeus, Allergan, Allgenesis, Alzheon, Amaros, Amgen, Annexon Biosciences, Apellis, AsclepiX, Astellas, Augen Therapeutics, Aviceda, AviadoBio, Bausch and Lomb, Bayer, Beacon Therapeutics (AGTC), Biocryst, Biogen, Bionic Vision Technologies, Boehringer Ingelheim, BVI, Carl Zeiss Meditec, Celltrion Healthcare Co., Clearside Biomedical, Complement Therapeutics, DORC, Endogena Therapeutics, EyeBiotech, Eyebiotech, EyePoint Pharmaceuticals, Frontera Therapeutics, Galimedix, Genentech, InGel, Innovent, Invirsa, iRenix, Isarna, Iveric Bio, Janssen Pharmaceuticals, jCyte, Kanaph Therapeutics, Kanghong, Kera Therapeutics, Kiora, Kodiak, Kriya Therapeutics, Merck, Nanoscope Therapeutics, Neurotech, Novartis, Novo Nordisk, Ocugenix, Ocuphire, Ocular Therapeutix, Oculis, OcuTerra, Omeros, ONL, Opthea, Optos, Osanni Bio, Oxurion, Panther Pharmaceuticals, Perceive Bio, Plano, Ray Therapeutics, Regeneron, RegenXBio, Resonance Medicine Inc., RetinaAI Medical AG, Retinal Sciences, ReVana, Revopsis, Ripple Therapeutics, Roche, Roivant, Samsung Bioepis, Sandoz, Sanofi, Santen, SGN Nanopharma Inc., Shanghai Henlius, Sitala, SmileBiotek Zhuhai Ltd, Stealth Biotherapeutics, Stuart, Sustained Nano Systems, Takeda, Théa, Tilak, Unity Biotechnology, Valo, Vanotech, and VisgenX.

Please respond by return email with an updated Funding Statement and Competing Interests Statement and we will change the online submission form on your behalf.
---

## [Editor Report · Acceptance letter]

PONE-D-25-59787

PLOS One

Dear Dr. Chaudhary,

I'm pleased to inform you that your manuscript has been deemed suitable for publication in PLOS One. Congratulations! Your manuscript is now being handed over to our production team.

Kind regards,

on behalf of

Dr. Andrzej Grzybowski

Academic Editor

PLOS One